# Assessment of Handler Exposure to Pesticides from Stretcher-Type Power Sprayers in Orchards

**Zhinan Wang** [1,†]**, Yuxi Meng** [1,2,†]**, Xiangdong Mei** [1]**, Jun Ning** [1]**, Xiaodong Ma** [2,*] **and Dongmei She** [1,*]

1    State Key Laboratory for Biology of Plant Diseases and Insect Pests, Institute of Plant Protection, Chinese Academy of Agricultural Sciences, No.2 Yuanmingyuan West Road, Beijing 100193, China; 82101182339@caas.cn (Z.W.); myxiixi@cau.edu.cn (Y.M.); xdmei@ippcaas.cn (X.M.); jning@ippcaas.cn (J.N.)

2    College of Science, China Agricultural University, Beijing 100193, China

*    Correspondence: dongxm@cau.edu.cn (X.M.); shedongmei@caas.cn (D.S.)

†    Zhinan Wang and Yuxi Meng are co-first authors and contributed equally to this work.

**Abstract:** The production and export volume of fruits from China are among the top three in the world. Pesticides are applied to orchards more than 10 times a year to control pests, and stretcher-type power sprayers are widely used to apply chemical pesticides. However, an assessment of pesticide-handler exposure to pesticides in this scenario has not been reported in China. The test pesticide, 30% SYP-9625 concentrate diluted 3000 times, was sprayed on apple orchards in Beijing China. Experiments were conducted to assess dermal and inhalation exposure using standard whole-body dosimetry and air-sampling methodologies. The dermal deposition was the main route of exposure in this study. The dermal unit exposure (UE) of handlers was 350 mg·kg$^{-1}$ a.i. of SYP-9625. The hands accounted for 59% of the total exposure and were the most exposed body part. Inhalation UE was 0.720 mg·kg$^{-1}$ a.i. of SYP-9625 and was negligible compared with dermal exposure. We found that use of protective garments while using stretcher-type powers sprayers reduced dermal pesticide exposure. These results can be used as a reference for the handler's safety in pesticide management and orchard mechanical management.

**Keywords:** inhalation exposure; dermal exposure; stretcher-type power sprayers; orchards

## 1. Introduction

While spraying crops, handlers are inevitably exposed to pesticides. Pesticide-handlers are directly exposed to pesticide clouds, allowing pesticides to readily enter the body through the skin and respiratory tract, which may result in acute poisoning and other chronic health issues [1–3]. Therefore, it is necessary to evaluate occupational exposure to pesticides to reduce the degree of exposure and protect handlers [4,5].

Women and the elderly, the main labor force in rural China, are not well educated and have a low level of security awareness [6]. Exposure to pesticides is one of the most serious occupational risks faced by them. Therefore, it is imperative to carry out exposure assessments for pesticides according to the fundamental realities of China.

In recent years, China has gradually paid attention to the occupational exposure assessment of pesticides. Now our exposure assessment is based on the methods used in developed countries in the West. The methods used by different people converge and there are certain differences. Dermal exposure is a key component of pesticide risk assessment and whole-body dosimetry is more accurate in estimating dermal exposure [7,8]. Yang et al. [9] and Gao et al. [10] used the whole-body sampling method to study the exposure of farmers using the 16-type knapsack manual sprayer to

spray chlorpyrifos in cornfields. Cao et al. [11] and Chen [12] studied the exposure of farmers to imidacloprid, chlorpyrifos, and lambda-cyhalothrin in wheat fields using whole-body dosimetry.

In 2018, orchards covered 11,875 thousand hectares in China, and fruit output reached 226.88 million tons [13]. The fruits covering the largest area were apples, citrus, and pears and the output of these fruits accounted for 37.6% of the total fruit production in the country [13]. Therefore, it is necessary to assess the exposure risk of operators to pesticides during treatment in apple orchards under defined-use scenarios.

In 1981, Franklin et al. [14] used air monitoring and patch techniques to estimate exposure, proposing an exposure measurement method for orchard application scenarios. Moon and Kim studied the pesticide exposure assessment of the application of fenvalerate and methomyl in apple orchards in Korea [15]. Thouvenin selected insecticide foliar application to a vineyard as the exposure scenario [16]. However, relevant reports on the occupational exposure to pesticides in orchards have not been studied in China.

In developed countries in Europe or North America, the application of pesticide chemicals is very specialized, serialized, and standardized. Depending on the application, each scene has a dedicated application machine [17]. China's pesticide application equipment models are aging, and the market share of manual sprayers reaches 80% [18]. Stretcher-type sprayers are mainly used for fruit-tree applications. This type of sprayer originated in the 1950s and 1960s and was developed to prevent serious injury from rice borers. The structure of the stretcher-type power sprayer consists of five parts—the frame, the power machine, the liquid pump, the water absorption part, and the spray part [19]. The diameter of the droplet is about 400–600 microns [20]. It has a long range and a wide spray coverage and is easy to use, but the pesticide utilization rate is only about 15% [20].

The pesticide SYP-9625 is a broad-spectrum acrylonitrile acaricide that is quick-acting and long-lasting [21]. As a newly-developed acaricide, no studies have been published on exposure risk assessment. The purpose of this work was to study the exposure data of occupational handlers using stretcher-type power sprayers for fruit tree application scenarios.

## 2. Materials and Methods

### 2.1. Reagents and Materials

Analytical standard SYP-9625 (CAS No., 1253429-01-4) (98% purity) was purchased from Shenyang Sinochem Agrochemicals R&D Co., Ltd. (Shenyang, China). The Structure of SYP-9625 is in Figure S1. The commercial SYP-9625 formulation used in the field trial was 30% suspension concentrate. Acetone and acetonitrile used for the extraction were of analytical grade; the acetonitrile used for the preparation of standard solutions was of high-performance liquid chromatography grade (Sigma-Aldrich, Steinheim, Germany). Lidded glass jars of 500 mL, 1000 mL, and 2500 mL sizes were used to extract samples. Cotton clothing was obtained from the Evolu Flagship Store (Hangzhou, China).

### 2.2. Field Trial

Operators were local farmers; they were also volunteers. They were experienced in the application of pesticides and in good health. They had agreed to the experimental procedures and co-operated in this study. Each operator was provided with a full explanation of the study and its requirement, and any potential risks. They could withdraw from the study at any time, and for any reason. A signed, informed consent form was obtained from each operator prior to his/her participation in the study.

The exposure study was conducted in Changping district, Beijing, China. At noon in summer, the temperature is too high to spray pesticide, and it can also cause phytotoxicity to plants. Farmers generally use pesticides in the morning or evening. So our experiments were conducted from 6:30 to 10:00 on 7 June 2017. The temperature was 17.5 to 37 °C, sunny with a relative humidity of 27–87%, and no wind. The experiment included four male handlers (1a, 1b, 2a, 2b, 3a, 3b, 4) with a total of seven exposures. The handlers were approximately 50 years old, with height and weight of

165–175 cm and 60–70 kg, respectively. The apple trees were in the fruiting stage, and their average height was 3 m. The rows were separated by 2.5–3.8 m, with 1.5–2.5 m between trees in the same row.

For pesticide application, the spray solution was prepared by mixing 300 L of water with 100 mL 30% suspension concentrate. Each handler followed their normal pace to spray pesticide across one acre. The application was carried out simultaneously by four stretcher-type electric sprayers. The machinery was provided by the farmers who planted the fruit trees and had been in use for 1 to 4 years. New, old, or different sprayers may have caused different exposures. The sprayer in Figure 1 is a relatively-new stretcher-type electric sprayer. Four farmers had more than ten years of experience in spraying pesticides. They randomly-selected instruments and used them to complete seven application experiments. In the process of application, the handlers sprayed according to their usual spraying habit in order to obtain real exposure. Their spray habits had been formed through long-term work. These farmers' spraying techniques were considered representative of typical spraying behaviors.

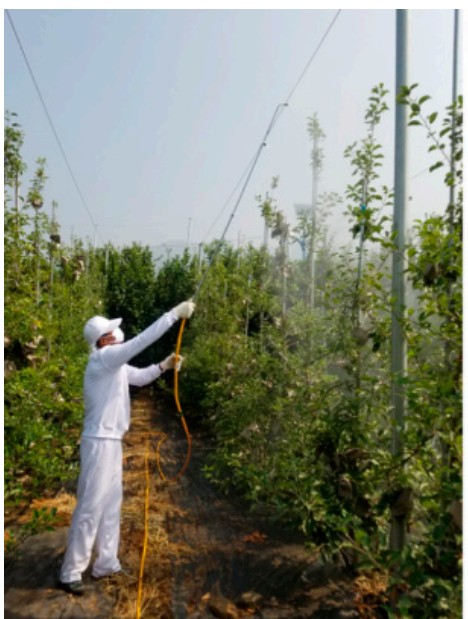 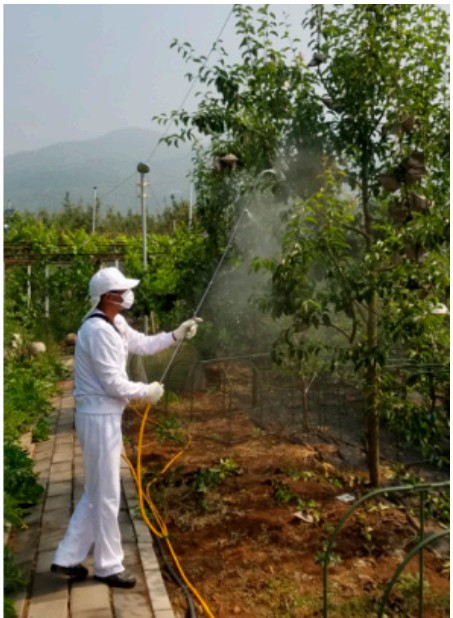

**Figure 1.** Stretcher-type power sprayer.

*2.3. Potential Dermal and Inhalation Exposure Monitoring*

2.3.1. Potential Dermal Exposure Monitoring

Dermal exposure was determined using a whole-body dosimetry method, including the two sets of clothing, inside and outside hat, masks, and inside and outside gloves. The basic requirements were as follows: (1) inner clothing—using a cotton content greater than 70%, white, thin, long-sleeved shirt and trousers, round neck, cuffs, neckline tightened; (2) outer clothing—generally with cotton content greater than 70%, white, thick, long-sleeved shirt and trousers, round neck, cuffs, neckline tightened; (3) inside hat—with eight layers of white gauze (20 × 40 cm); (4) outside hat—single-layer, cotton white hat with brim; (5) masks—medical gauze masks; (6) inner gloves—white thin glove with a cotton content greater than 70%; and (7) outer gloves—white thick-lined gloves with cotton content greater than 70%. For accurate quantitative calculations, the handlers wore two layers of clothing during application, and the inner layer of clothing was used to simulate the skin.

2.3.2. Potential Inhalation Exposure Monitoring

Inhalation exposure was measured using a personal air monitor equipped with a portable battery-operated sampling pump and a solid sorbent tube (ORBO 609 Amberlite® XAD-2 400/200 mg). XAD-2 resin was used for capturing the pesticides in the air (each handler's breathing zone). Personal air

sampling pumps with XAD-2 filter tubes were used to monitor inhalation exposure. Inhalation UE was calculated with the formula:

$$UE(mg \cdot kg^{-1} \text{ of a.i.handled}) = \frac{\text{exposure dose(mg)} \times \text{application time(min)} \times \text{air change rate}(L \text{ min}^{-1})}{\text{flow rate}(L \text{ min}^{-1}) \times \text{sampling time(min)} \times \text{kg of a.i. handled}}$$

An air change rate of 29 L·min$^{-1}$ and an air sampler's pumping rate of 2 L·min$^{-1}$ was assumed [22]. The application and sampling time was assumed to be equal because the orchard size was small and there was no need for rest periods during applications.

### 2.4. Handler Sampling

After the end of the application, the air sampling pump was turned off and the XAD-2 tubes were collected for cryopreservation and transportation. Sampling time was defined as the time from the start to the end of spraying. The protective clothing was divided into six parts as depicted in Figure 2, labeled, and transported in the dark to be stored under freezing conditions.

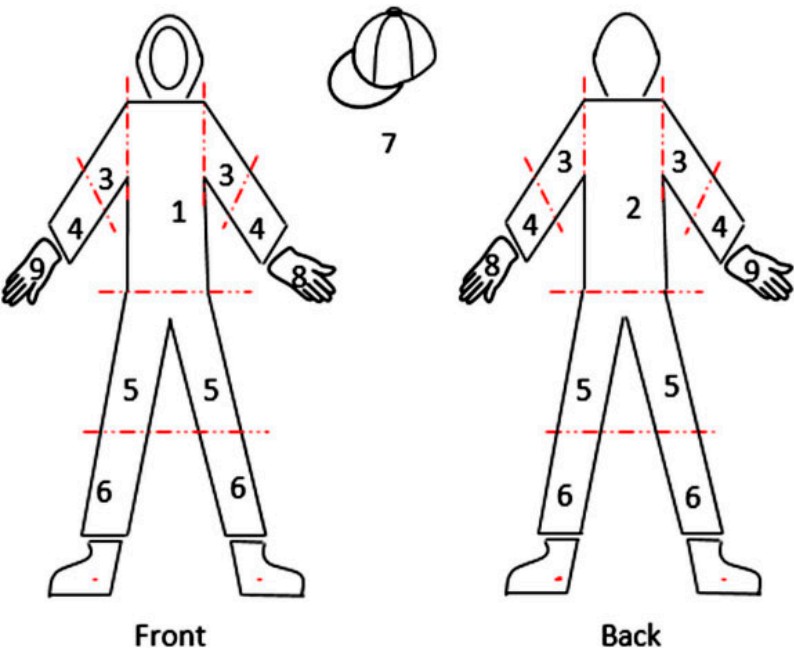

**Figure 2.** Garment sectioning for whole-body analysis.

Some body parts were cleaned and the resultant wash was collected, as follows: the assistant wore clean disposable gloves and thoroughly wiped the worker's face/neck with moist medical gauze. Approximately 4 mL of 0.01% Aerosol OT(Sodium dioctyl sulfosuccinate) was evenly distributed over the gauze. The steps above were repeated. Each gauze was then collected and labeled. The handlers immersed both hands in 400 mL of 0.01% Aerosol OT solution and scrubbed carefully for at least 30 s. The hands were then rinsed with about 100 mL of solution, which was consistent with the above. This 500 mL sample was collected and labeled [23].

To evaluate the stability of SYP-9625 during storage and transportation, an on-site addition recovery test was carried out in the area near the application site. A certain concentration of pesticide standard solution was prepared, the pesticide solution was spread evenly on a complete part of the clothes (such as thighs), and the volume recorded. Then the clothes were put in a sealed bag and transported under the same conditions as the experimental samples. Each material had a different amount of additive, and two repetitions were taken: (1) underwear (size: 30 cm × 30 cm) 10× and 100× LOQ (limit of quantitation); (2) outer coat (size: 30 cm × 30 cm) 100× and 1000× LOQ; (3) handwashing solution (solution volume: 50 mL) 10× and 100× LOQ; (4) facial and neck wipes 20× and 200× LOQ;

(5) inner gloves 10× and 100× LOQ; (6) outer gloves 100× and 1000× LOQ; and (7) air filter samples 10× and 100× LOQ.

The LOQ of each component and matrix reflected the minimum amount of data that could be quantified.

The samples from the on-site additional recovery test had the same environmental conditions as the samples from the application and used the same transport and storage conditions.

### 2.5. Chromatographic Conditions

SYP-9625 was analyzed using Ultra-performance liquid chromatography tandem mass spectrometry (UPLC-MS) (AQUITY TQD, Waters, Milford, MA, USA) with a C18 column (2.1 mm × 100 mm, 1.7 μm, Waters, UPLC® BEH). The mobile phase was acetonitrile (A) and 0.1% formic acid-water (B). The linear mobile phase gradient started at 40% A (0 to 0.5 min), increasing to 80% A (0.5 to 3.5 min), after which the column was equilibrated at 40% A (3.5 to 5.0 min). Tandem mass spectrometry was operated in positive electrospray ionization (ESI) and multiple reaction monitoring (MRM) mode. The optimized values of the MS parameters were as follows: source temperature 150 °C, desolvation temperature 350 °C, cone gas ($N^2$) flow of 50 $L \cdot h^{-1}$, desolvation gas ($N^2$) flow of 650 $L \cdot h^{-1}$. The analytical instrument control, data acquisition, and processing were performed by MassLynx V4.1 software (Waters). The mass spectrometric parameters for UPLC-MS determination of SYP-9625 concentrations were listed in Table S1. The UPLC-MS chromatogram for SYP-9625 were in Figure S2.

### 2.6. Linear Range

Linear regression was performed between 0.0014 to 1.4 $mg \cdot L^{-1}$ for SYP-9625. The derived calibration curve formula was y = 241643x + 4475.8 ($R^2$ = 0.9987). The limit of detection (LOD) of the compounds was calculated on a signal-to-noise (S/N) ratio of 3 concerning the background noise obtained from the blank sample, whereas the limit of quantitation (LOQ) was via an S/N ratio of 10. The LOD of SYP-9625 was 0.010 $ug \cdot L^{-1}$ and LOQ was 0.030 $ug \cdot L^{-1}$.

### 2.7. Extraction and Recovery of SYP-9625

Pesticide extraction methods for similar matrices have been reported in the literature, and the methods are briefly described as follows [23]: According to the size of different parts of the clothes, the protective suits were ultrasonically extracted with different volumes of acetone for 20 min at 25 °C. The operation was repeated once for better recovery. The volume of the extraction solvent was sufficient to immerse the clothing and was recorded in detail. The extract was then concentrated on a rotary evaporator and dissolved in 2 mL EtOAc(Ethyl acetate). The XAD-2 sample was extracted twice with 5 mL of acetone, then concentrated with a nitrogen evaporator and dissolved in 1 mL of acetonitrile. For 0.01% Aerosol OT, 10 mL of acetonitrile was added to a 10 mL sample and shaken vigorously, then NaCl was added to the mixture, shaken, and centrifuged at 3600 $r \cdot min^{-1}$ for 5 min to separate the organic solvent from the water. Finally, 1 mL of the resulting supernatant was directly injected into UPLC-MS with an injection volume of 10 uL.

### 2.8. Statistical Analysis

Results were expressed as mean ± SD and statistical significance was determined by one-way analysis of variance (ANOVA) followed by the least significant difference (LSD) using the statistical software SPSS (SPSS 19.0; SPSS Inc., Chicago, IL, USA). The level of statistical significance was established at $p < 0.05$.

## 3. Results and Discussion

### 3.1. Validation of the Analytical Method

Recovery experiments were carried out at three different fortification levels in nine replicates and the relative recovery rates were calculated via the matrix-matched calibration curves. Acetone was chosen as the extraction solvent for cotton wool, gauze, and XAD-2. In this study, SYP-9625 detected in different materials had a high recovery rate, and a method of detection and analysis of SYP-9625 concentrations in different matrices was established. Acceptable recoveries were obtained in the ranges of 91.02–102.35% from cotton wool, 97.34–103.45% from gauze, 95.33–107.26% from outer gloves, 93.28.9–97.65% from XAD-2, and 103.25–110.37% from 0.01% Aerosol OT. The recovery and precision of SYP-9625 (expressed as relative standard deviation) are shown in Table 1, RSDs were less than 10% in all samples, demonstrating the excellent repeatability of the process. Therefore, a combination of UPLC-MS detection, and the utilized extraction method could serve as a conventional detection method for SYP-9625 in all of the above matrices.

**Table 1.** SYP-9625 recovery from field-fortified samples.

| Material | Fortification Concentration (mg·kg$^{-1}$) | Recovery (%) | RSD (%) |
|---|---|---|---|
| Cotton clothing | 0.1 | 97.28 | 2.8 |
| | 0.2 | 91.02 | 3.1 |
| | 1 | 102.35 | 3.4 |
| Gauze | 0.01 | 101.00 | 2.0 |
| | 0.02 | 103.45 | 2.6 |
| | 0.1 | 97.34 | 1.8 |
| Outside gloves | 0.05 | 95.33 | 4.7 |
| | 0.1 | 107.26 | 3.6 |
| | 0.2 | 105.15 | 5.3 |
| XAD-2 | 0.01 | 94.84 | 3.2 |
| | 0.02 | 97.65 | 1.9 |
| | 0.04 | 93.28 | 2.8 |
| 0.01% Aerosol OT | 0.025 | 110.37 | 2.2 |
| | 0.05 | 106.00 | 1.6 |
| | 0.1 | 103.25 | 3.8 |

Since this orchard is far away from our laboratory, we needed to clarify whether the pesticides on the various exposed substrates would be lost during transportation The recovery rate of the sample added to the on-site recovery test was approximately 91.5%. This proves that the loss of pesticides during storage and transportation is below 10%. The OECD(Organization for Economic Co-operation and Development) pointed out that the samples were sufficiently stable within 30% decline of the recovery rate [24].

### 3.2. Dermal and Inhalation Exposure during Application

In this study, we measured pesticide exposure by dermal and inhalation routes using standard systemic dosimetry and air sampling methods. In the practical application of pesticides, farmers wear clothes with at least one layer of clothing. Actual dermal exposure (ADE) is the amount of pesticide that passes through the clothes and is exposed to the skin. The handlers wear two layers of clothing during application and the inner layer of clothing is used to simulate the skin. Potential dermal exposure (PDE) during application is the sum of dosimeter readings for the inner and outer layers. The pesticide test results for the inner clothing, hand wash, and face/neck wipes were calculated as actual dermal exposure (ADE) [23]. The inhalation exposure was calculated as the amount of active compound adsorbed by the XAD-2 sorbent tube on the air sampler throughout the application period. The pesticide exposure in this study was measured by PDE. Table 2 lists the pesticide unit exposures of the two layers of clothing inside and outside the applicator. According to the literature,

the occupational exposure of pesticides in the field is quite different for different pesticide application personnel, and the difference is related to the operating habits and proficiency [10]. Therefore, the final experimental results are based on average.

**Table 2.** Unit exposures at various parts of the body (mg/kg a.i. handled).

| Part of Garment | 1 | 2 | 3 | 4 | 5 | 6 | 7 | Mean | SD |
|---|---|---|---|---|---|---|---|---|---|
| a1 | 0.10 | 0.18 | 0.32 | 0.07 | 0.09 | 0.16 | 0.12 | 0.15 | 0.08 |
| a2 | 0.23 | 0.18 | 0.08 | 0.08 | 0.06 | 0.08 | 0.06 | 0.11 | 0.07 |
| a3 | 0.07 | 0.07 | 0.02 | 0.11 | 0.66 | 0.02 | 0.01 | 0.14 | 0.23 |
| a4 | 0.02 | 0.57 | 0.05 | 0.05 | 0.03 | 0.05 | 0.09 | 0.12 | 0.20 |
| a5 | 0.18 | 0.23 | 0.14 | 0.05 | 0.25 | 0.08 | 0.13 | 0.15 | 0.07 |
| a6 | 0.15 | 0.92 | 0.12 | 0.30 | 0.10 | 0.02 | 0.16 | 0.25 | 0.31 |
| a7 | 0.78 | 0.26 | 0.13 | 0.14 | 1.23 | 2.58 | 4.03 | 1.31 | 1.48 |
| a8 | 0.98 | 99.48 | 7.43 | 0.22 | 0.17 | 2.35 | 53.78 | 23.49 | 38.71 |
| a9 | 0.31 | 124.26 | 9.27 | 0.25 | 0.02 | 2.63 | 40.19 | 25.28 | 45.97 |
| b1 | 44.18 | 31.01 | 4.92 | 6.76 | 3.75 | 17.83 | 18.46 | 18.13 | 15.03 |
| b2 | 7.07 | 5.54 | 4.56 | 4.64 | 4.28 | 42.70 | 16.04 | 12.12 | 14.11 |
| b3 | 4.87 | 14.36 | 2.50 | 3.71 | 4.85 | 21.38 | 15.53 | 9.60 | 7.38 |
| b4 | 4.12 | 47.16 | 5.02 | 6.59 | 4.57 | 14.92 | 33.84 | 16.60 | 17.17 |
| b5 | 11.41 | 20.11 | 11.32 | 0.04 | 3.95 | 22.07 | 28.13 | 13.86 | 10.10 |
| b6 | 66.30 | 98.75 | 135.68 | 0.08 | 7.81 | 42.49 | 106.29 | 65.34 | 51.37 |
| b7 | 2.52 | 9.39 | 0.58 | 0.26 | 4.96 | 10.91 | 16.13 | 6.39 | 5.95 |
| b8 | 35.95 | 253.94 | 59.09 | 17.15 | 32.62 | 86.76 | 128.87 | 87.77 | 82.50 |
| b9 | 16.12 | 234.01 | 23.58 | 31.48 | 9.82 | 91.44 | 72.15 | 68.37 | 79.07 |
| 10 | 1.33 | 0.26 | 0.21 | 0.12 | 0.29 | 0.38 | 2.33 | 0.70 | 0.83 |
| 11 | 0.16 | 0.05 | 0.01 | 0.02 | 0.02 | 0.19 | 0.64 | 0.16 | 0.23 |
| 12 | 0.31 | 0.06 | 0.15 | 0.02 | 0.02 | 0.20 | 0.59 | 0.19 | 0.20 |
| 13 | 0.03 | 0.14 | 0.02 | 0.01 | 0.02 | 0.02 | 0.11 | 0.05 | 0.05 |
| 14 | 0.009 | 0.005 | 0.01 | 0.004 | 0.048 | 0.02 | 0.04 | 0.02 | 0.02 |

A, inner-layer garment; b, outer-layer garment; 1, front torso (above the waist); 2, rear torso (above the waist); 3, right and left upper arms (shoulder to elbow); 4, right and left forearms (elbow to cuff); 5, right and left thighs (waist to knee); 6, right and left lower shins (knee to cuff); 7, cap; 8, left glove; 9, right glove; 10, mask; 11, face wipe; 12, neck wipe; 13, hand washes; 14, XAD-2.

### 3.2.1. Dermal Unit Exposure during Application

The total PDE of handlers was 350 mg·kg$^{-1}$. These data is close to the exposure assessment data for the fruit tree application scenario reported by Zhao et al. [25] and lower than the exposure data for the knapsack sprayer in cornfields and peanut fields reported by Gao et al. [10] and Chen et al. [12], but higher than the exposure data using pesticide speed sprayer application in an apple orchard [26]. As shown in Table 3, according to the UE of each garment, the highest contaminated sections were hands and shins, accounting for an average of 59% and 19% of total dermal exposure, respectively.

**Table 3.** Distribution of dermal exposure during application (mg/kg a.i. handled).

| | Exposure Levels of ADE | Total Exposure Levels of PDE |
|---|---|---|
| Front torso | 0.15 ± 0.08b | 18.13 ± 15.03a |
| Rear torso | 0.11 ± 0.07b | 12.12 ± 14.11a |
| Upper arms | 0.14 ± 0.23b | 9.60 ± 7.38a |
| Forearms | 0.12 ± 0.20b | 16.60 ± 17.17a |
| Thighs | 0.15 ± 0.07b | 13.86 ± 10.10a |
| Shins | 0.25 ± 0.31b | 65.34 ± 51.37a |
| Hands | 48.82 ± 84.21a | 204.96 ± 241.57a |
| Head | 1.31 ± 1.48b | 8.75 ± 8.17a |
| Total | 51.05 ± 84.70b | 350.28 ± 306.97a |

All data are presented as mean ± standard error (n = 7). Different lessters(a, b) in each row indicate significant difference at $p \leq 0.05$ (least significant difference).

We found that the exposure of hands, which refers to the sum of the exposure of the left and right gloves plus that of the handwashing solution, was considerably greater than other body parts. This is different from the results of Zhao et al. [15,24]. While Li et al. [27] showed that when spraying high places with a spray gun, the most contaminated sections were hands. This could be attributed to the different types of application equipment used. In our experiments, the applicator needed two hands to hold the sprayer in order to work properly, and the hands were the only body part in direct contact with the applicator. High pressure can cause pesticide liquid overflow in the connection between the hose and spray lance onto the hand. Further, most of the pesticide liquid sprayed from the nozzle, but liquid flow along the spray lance to the hand was possible. In addition, any problems that occurred during the application process needed to be addressed using both hands, such as pulling a hose for pesticide delivery, and handling the leakage of pesticide [28,29]. The high exposure level of shins may be due to a large number of pesticides deposited on the ground weeds during the application of pesticides into the base of apple trees [30,31]. In addition, handlers needed to walk through the weeds in order to facilitate the application of pesticides [25]. These results are similar to those obtained by Noh et al. [32], who reported when spraying onto lower crops (about 80–100 cm high), workers are exposed while moving.

However, when handlers are wearing long pants and pure cotton gloves, hand exposure decreased by 76%, shin exposure decreased more than 99%, and the ADE was reduced by 85.43% to 51.1 mg·kg$^{-1}$. This shows that clothing plays a very good role in protecting against pesticides, especially for those who regularly apply pesticides. This is consistent with the conclusions of Ren et al. [33] and An et al. [34]. Hands were still exposed significantly, so the protection of the hands during the application of pesticides should be given extra attention. In order to further reduce the contamination of the hands, we recommend that handlers wear impermeable gloves such as chemical-resistant gloves. The exposure of the left and right hand is approximately the same (Table 4), because during the application process, the handlers need to change the way in which the spray gun is held based on the direction in which they are walking, and the height of the fruit tree, in order to facilitate spraying.

**Table 4.** Exposure levels of hands (mg/kg a.i. handled).

|  | ADE of Hands | PDE of Hands |
| --- | --- | --- |
| Left hands | 23.49 ± 38.71a | 111.26 ± 120.23a |
| Right hands | 25.28 ± 45.97a | 93.61 ± 123.10a |

All data are presented as mean ± standard error (n = 7). Letter (a) in each column indicate significant a difference at $p \leq 0.05$ (least significant difference).

In the orchard application scenario, a handler stands under the tree to apply the pesticide to the oblique crown above the tree. The liquid droplets collide with the leaves and bounce to fall under the force of gravity [35]. The pesticide drifts to the head and body of the handler. During our experiment, the handler's head was protected by a cotton baseball cap. The exposure to the head was not significant, at only 8.75 mg·kg$^{-1}$, which may be related to the small overall head area. As can be seen from Table 3, the wearing of a cap reduced the skin exposure of the head to 1.31 mg·kg$^{-1}$, and the protection rate was about 85%. However this protection rate was relatively low compared with other body parts. Since the brain is an essential organ that controls many human organs, the head should be better protected. Therefore, we recommend wearing a wide-brimmed sun visor with cotton or waterproof material during the actual application of pesticides. This can increase the coverage of the head and neck to prevent the pesticide liquid from scattering on the head. At the same time, it can also protect the skin from sunlight and reduce the damage of ultraviolet rays to the skin. In fact, farmers usually wear straw hats when they work. This kind of hat can prevent pesticides from falling on the face and neck. This protective measure can be considered in future experiments.

### 3.2.2. Inhalation Exposure during Application

Inhalation exposure occurred when airborne pesticide vapors or droplets appeared in working areas owing to the application of pesticides.

In this application scenario, the inhalation unit exposure was 0.72 mg·kg$^{-1}$, which was only 0.13% of the total skin exposure. The data were slightly higher than the orchard pesticide application scenario reported in Korea ($0.7 \times 10^{-3}$ mg) [25]. This may be because our fruit trees are denser, the workers are close to the fruit trees during the application process, and the whole process of application is under the cloud of pesticides, therefore more pesticides are inhaled.

Stretcher-type electric sprayers are widely used in orchards. Therefore, more experimental data on citrus, grape, pear, and other fruits are needed to get diverse results, also a larger sample size would be good for representative results.

## 4. Conclusions

This occupational exposure assessment studied the application scenario of a stretcher-type electric sprayer, using the apple tree as an example. This is a typical Chinese orchard application scenario. This is the first time a study has focused on occupational pesticide exposure in orchards in China. The data from this study are a valuable guide for the application of pesticides on fruit trees such as citrus, pear, peach, and other small shrub type fruit trees.

In this study, we performed a systematic occupational exposure assessment on stretcher-type electric sprayer application of SYP-9625 in Chinese orchard fields using standard whole-body dosimetry and air sampling methodologies. We also verified the method of extracting and detecting SYP-9625 from various matrix materials. The results indicate that the hand is the most exposed body part and needs special protection. However, the total exposure is at the same level as that shown in other countries' research. One layer of clothing could protect from dermal exposure, causing a reduction of approximately 86%. Therefore, long trousers and waterproof gloves are necessary to protect the health and safety of the handler. The stretcher-type electric sprayer is China's most important orchard application equipment, but its related occupational exposure data are scarce. Therefore, it is necessary to strengthen the data in this research area.

Our experiment was only carried out in Beijing, and more studies should be performed to obtain extensive experimental results in the future. Our data might help establish accurate and strategic predictive exposure models and databases for risk assessment and pesticide registration.

**Supplementary Materials:** The following are available online at http://www.mdpi.com/2076-3417/10/23/8684/s1, Table S1: Mass spectrometric parameters for UPLC-MS determination of SYP-9625 concentrations, Figure S1: the structure of SYP-9625, Figure S2: the UPLC-MS chromatogram for SYP-9625.

**Author Contributions:** Experimental design, Z.W. and D.S.; investigation, Z.W. and Y.M.; data analysis, Z.W. and Y.M.; writing—original draft, Z.W. and Y.M.; writing—review and editing, X.M. (Xiangdong Mei) and D.S.; funding acquisition, D.S. and X.M. (Xiaodong Ma); supervision, J.N. All authors have read and agreed to the published version of the manuscript.

**Funding:** This research was funded by The National Key Research and Development Program of China (grant number 2016YFD0200201), the National Natural Science Foundation of China (grant number 31772175), and the National Natural Science Foundation of China (grant number 31621064).

**Conflicts of Interest:** The authors declare no conflict of interest.

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
