# Peer review of "Assessment of Handler Exposure to Pesticides from Stretcher-Type Power Sprayers in Orchards"

_applsci, doi:10.3390/app10238684_

Round 1

Reviewer 1 Report

The present research reports the occupational exposure to SYP-9625 pesticide on fruit trees in orchards in China. This reviewer highlights that the analytical method for determination of SYP-9625 pesticide was properly validated. The manuscript fits within the scope of the journal and results can be considered of interest, the manuscript is well contextualized and properly discussed. Thus, this reviewer have only some minor comments:

  • Line 59. “machine15” should be replaced by “machine[15]”.
  • Line 60. “sprayers16” should be replaced by “sprayers[16]”.
  • Line 279-284. The paragraph “Personal air sampling pump ………… periods during applications.” should be relocated at subsection 2.3.2.

Author Response

Dear Reviewers:

We are very grateful to you for reviewing the paper so carefully and we have tried our best to improve the manuscript and make some changes.

Here’re the responses to your comments:

Line 59. “machine15” should be replaced by “machine[15]”.

We are very sorry for our incorrect writing and the format and content of the references have been corrected at line 58.

Line 60. “sprayers16” should be replaced by “sprayers[16]”.

It is really a good idea as you suggested, and we have changed them all to meet your thoughts.at line 59.

Line 279-284. The paragraph “Personal air sampling pump ………… periods during applications.” should be relocated at subsection 2.3.2.

It is indeed more appropriate to migrate this paragraph to 2.3.2, and we have put it in lines 121-126.

Thanks again for your help!

Reviewer 2 Report

The overall quality of the manuscript is rather good. The paper is well written and I recommend to the authors to continue working on these topics since it represents an important health matter for rural workers.

Author Response

Dear Reviewers:

We are very grateful to you for reviewing the paper so carefully and thank you for your recognition of our work.

Best wishes!

Reviewer 3 Report

SYP-9625 is a rather novel acaricide, widely used in fruit growing, not only in China. Studies on the risk us users of SYP-9625 are not yet available. Authors investigated the exposure of handlers to the pesticide during spraying in orchards. The study is solid, but scientifically not much exciting. Main results are an about 60% exposure of the hands during spraying, whereas inhalation of the pesticide was low. In spite of the limited scientific value of the paper, I suggest to accept the manuscript for Applied Sciences after thorough editorial and language revision.

Specifics:

  • line 17 and others: change dermal etc. in lowercase letters
  • line 41: this is not a correct way of citation
  • line 44 and others: give reference numbers in square brackets and without a comma before the bracket
  • line 66: delete the comma after the pesticide name
  • line 99 and others: delete the full stop before figure numbers
  • line 177: give a g value for the centrifugation step
  • line 225: the first sentence is not complete
  • line 251: do authors mean skin exposure?
  • line 253: again, the citation is not correct
  • References have to by carefully checked (Nicaraguan, China etc. in uppercase letters; use correct acronyms for journal titles; all species names in italics)

Author Response

Dear Reviewers:

We are very grateful to you for reviewing the paper so carefully and we have carefully considered the suggestion and make some changes.

line 17 and others: change dermal etc. in lowercase letters

We are very sorry for our incorrect writing and the case has been corrected at line 17.

line 41: this is not a correct way of citation

We are very sorry for our incorrect citations. We have checked the citations carefully and the modified part has been marked in red.

line 44 and others: give reference numbers in square brackets and without a comma before the bracket

We are very sorry for our incorrect writing. We have put the reference number in square brackets and deleted the comma before the brackets.

line 66: delete the comma after the pesticide name

We are very sorry for our incorrect writing. The comma after the pesticide name has been deleted in line 65.

line 99 and others: delete the full stop before figure numbers

We are very sorry for our incorrect writing. The full stop before (Fig.1) has been deleted.

line 177: give a g value for the centrifugation step

We are very sorry for our unclear presentation and the g value and time for the centrifugation step has been added in line 180.

line 225: the first sentence is not complete

PDE means the potential dermal exposure and this abbreviation has been defined at line 214. So we use PDE to express potential dermal exposure and make some changes to express the meaning of the sentence more clearly

line 251: do authors mean skin exposure?

The ADE means the actual dermal exposure and this abbreviation has been defined at line 216.

line 253: again, the citation is not correct

We are very sorry for our incorrect citations. We have checked the citations carefully and the modified part has been marked in red.

References have to by carefully checked (Nicaraguan, China etc. in uppercase letters; use correct acronyms for journal titles; all species names in italics)

We are very sorry for our incorrect writing. We have made some changes, checked the citations carefully.

Thank you again for your valuable comments on our work!
